# High Prevalence of Metabolic-Associated Fatty Liver Disease (MAFLD) in Children and Adolescents with Severe Obesity

**DOI:** 10.3390/jcm14103565

**Published:** 2025-05-20

**Authors:** Magdalena Mierzwa, Żaneta Malczyk, Mirosław Bik-Multanowski, Stephanie Brandt-Heunemann, Bertram Flehmig, Ewa Małecka-Tendera, Artur Mazur, Elżbieta Petriczko, Michael B. Ranke, Martin Wabitsch, Małgorzata Wójcik, Agata Domżol, Agnieszka Zachurzok

**Affiliations:** 1Department of Pediatrics, Faculty of Medical Sciences in Zabrze, Medical University of Silesia, 41-800 Zabrze, Polandadomzol@sum.edu.pl (A.D.);; 2Department of Paediatric Endocrinology, University Hospital No 1, Medical University of Silesia in Katowice, 3 Maja St. 13-15, 40-055 Katowice, Poland; 3Department of Medical Genetics, Faculty of Medicine, Jagiellonian University Medical College, 31-007 Cracow, Poland; 4Institute of Human Genetics, University Hospital, LMU Munich, 80336 Munich, Germany; 5Center for Rare Endocrine Diseases, Division of Pediatric Endocrinology and Diabetes, Department of Pediatrics and Adolescent Medicine, 89075 Ulm, Germany; 6Mediagnost GmbH, 72770 Reutlingen, Germany; 7Department of Pediatrics and Pediatric Endocrinology, School of Medicine in Katowice, Medical University of Silesia, 40-055 Katowice, Poland; 8Department of Pediatrics, Pediatric Endocrinology and Diabetes, Medical Faculty, University of Rzeszów, 35-959 Rzeszów, Poland; drmazur@poczta.onet.pl; 9Department of Pediatrics, Endocrinology, Diabetology, Metabolic Disorders and Cardiology of Developmental Age, Pomeranian Medical University, 70-204 Szczecin, Poland; 10University Children’s Hospital, 72070 Tübingen, Germany; 11Department of Pediatric and Adolescent Endocrinology, Pediatric Institute, Jagiellonian University Medical College, 31-007 Cracow, Poland

**Keywords:** metabolic profile, children, adolescents, severe obesity, metabolic fatty liver disease

## Abstract

**Background**: Severe obesity in children and adolescents presents a particular health burden due to high prevalence of complications and comorbidities, including metabolic-associated fatty liver disease (MAFLD). The aim of this study was to assess the prevalence of MAFLD in Polish children and adolescents with severe obesity, and assess its relation to anthropometric profiles and metabolic risk factors. **Patients and Methods**: In 212 children and adolescents with severe obesity (aged 3–18 years), physical examination, body composition, liver ultrasound (US), and biochemical assessment were performed. MAFLD was diagnosed based on the presence of steatosis in US and/or persistently elevated alanine aminotransferase concentration. **Results**: MAFLD was present in 125 (59.0%) patients. Metabolic Syndrome (MetS) was diagnosed among 57.5% of patients without MAFLD, and in 66.5% of patients with MAFLD (*p* > 0.05). Impaired fasting glucose, impaired glucose tolerance, and high HOMA-IR occurred more frequently in MAFLD than in non-MAFLD patients (*p* < 0.05). In the MAFLD group, a higher number of patients with ≥4 components of MetS were observed than in the non-MAFLD group (33.6% vs. 12.6%, *p* = 0.0004). **Conclusions**: The prevalence of MAFLD among children and adolescents with severe obesity was 59%. These patients are characterised by more pronounced insulin resistance and higher number of MetS components.

## 1. Introduction

Data from 36 European countries from 2015 to 2017 showed that nearly 30% of children had excessive weight, and about 10% suffered from obesity [1]. A particular health burden, with a high prevalence of complications and comorbidities, is severe obesity, which is defined as BMI for age above +3 z-scores relative to the WHO growth reference median. Previous studies have shown that the prevalence of severe obesity is estimated as being from 1.0% to 5.5% in European children [2]. The prevalence of severe obesity among Polish children and adolescents remains unexplored to date, as unexplored are its comorbidities.

Obesity is associated with metabolic comorbidities such as hypertension, dyslipidemia, hyperinsulinemia, and type 2 diabetes (T2DM), and is a major risk factor for the development of fatty liver disease [3].

In 2020, the international consensus redefined adult fatty liver disease associated with metabolic dysregulation and proposed the term metabolic-associated fatty liver disease (MAFLD) as a new name for non-alcoholic fatty liver disease (NAFLD) [4]. Experts reached the consensus that the new definition takes into account the fundamental role of metabolic dysfunction in the pathogenesis of hepatosteatosis and reflects it in diagnostic criteria. Following this new nomenclature, the requirements for diagnosing MAFLD in children and adolescents were published in 2021 [3]. According to the International Paediatrics Gastroenterology Societies, NAFLD was a diagnosis of exclusion defined as chronic hepatic steatosis in children. In diagnosis, it is necessary to exclude other chronic liver diseases, mainly genetic, metabolic, infectious, or toxic causes [5]. Therefore, in clinical practice, it was challenging to identify NAFLD in children and adolescents. Misdiagnoses preclude appropriate prevention of long-term sequelae. A new term, MAFLD, is defined as fatty infiltration of the liver (more than 5% of hepatocytes) confirmed by imaging (liver ultrasonography, magnetic resonance, computer tomography), based on liver histology (reference standard) or blood biomarker [persistently elevated alanine aminotransferase (ALT) concentrations to more than twice the upper limit of normal for a gender], in addition to one of the three criteria: excess adiposity, presence of prediabetes or T2DM, or at least two evidence of metabolic risk abnormalities [increased waist circumference (WC), hypertension, hypertriglyceridemia, low HDL cholesterol (HDL-C) concentrations, impaired fasting glucose, and increased triglyceride-to-HDL-C ratio] [3]. The incidence of fatty liver disease is estimated at 3–10% in the paediatric population, up to about 40% in children and adolescents with obesity [3,6,7]. MAFLD is a liver disorder spectrum, including both simple hepatosteatosis and complications such as steatohepatitis (MASH) and cirrhosis. The pathophysiology of MAFLD is complex, containing genetic, epigenetic, environmental, metabolic, and inflammatory factors, and is still incompletely understood. Moreover, MAFLD is closely related to insulin resistance (IR) and dyslipidemia [3].

We hypothesised that MAFLD is strongly correlated with the presence of components of metabolic syndrome (MetS) in children and adolescents with severe obesity. Currently, there is limited research on the prevalence of MAFLD in Caucasian children with severe obesity. To complement these data, we conducted a study among Polish children and adolescents with severe obesity, comparing the risk factors of MetS between patients with MAFLD and patients without MAFLD (non-MAFLD). This analysis of data from the severe early-onset obesity (SEOO) cohort aims to evaluate the prevalence of MAFLD in a cohort of Polish children and adolescents with severe obesity, and assess the relationship between hepatosteatosis and anthropometric profiles and metabolic risk factors.

## 2. Materials and Methods

This study is part of a prospective, cross-sectional, multi-centre clinical study conducted in four specialised medical centres in Poland. It describes the preliminary results of the project, aiming to establish a Polish database of severely obese children and adolescents, to assess them clinically and biochemically, and to evaluate the prevalence of monogenic forms of obesity in this cohort [8].

The study is conducted according to the Declaration of Helsinki on “Ethical Principles for Medical Research in Humans” (9 July 2018). It was approved by the local Ethics Committees (No. PCN/CBN/0022/KB1/137/I/21/22, KBETUJ1072.6120.69.2022, KB-006/12/2022). At each participating institution, informed written consent was obtained from every patient’s parents/guardians, and from every patient over the age of 13.

### 2.1. Study Population

Two hundred and twelve patients (98 boys, 114 girls) between the ages of 3–18 years diagnosed with severe obesity in four Polish medical centres between May 2022 and September 2023 were recruited. The patients have met the following criteria:Inclusion criteria:1.Age 3–18 years.2.The presence of severe obesity defined as a BMI > 30 kg/m^2^ in children aged 3–6 years, BMI > 35 kg/m^2^ in children aged 6–14 years and BMI > 40 kg/m^2^ in children aged > 14 years [9].3.Hyperphagia and food-seeking behaviours.4.Written informed consent of the patient’s parent/guardian and patient above the age of 13 years to participate in the study.Exclusion criteria:1.Lack of written informed consent from the patient’s parent/guardian or patient above the age of 13 years.2.Secondary cause of obesity: previously diagnosed genetic syndrome coexisting with obesity, treatment with medicine with known effect on weight gain (glucocorticoids, valproic acid, risperidone, and others), Cushing’s syndrome, and other secondary causes of obesity.


In all patients, during a single visit to the clinical centre, the clinical, laboratory, and ultrasound data were obtained. The subjects underwent physical examination with anthropometric measurements. Body weight was measured to the nearest 0.1 kg on a calibrated balance beam scale, and body height was measured to the nearest 0.1 cm. WC was measured at the midpoint level between the lowest rib and iliac crest. In all children, the pubertal stage (development of breast, genitalia, and pubic hair) was assessed according to the classification by Tanner [10,11]. Blood pressure was measured with a calibrated automatic blood pressure monitor, with a cuff size appropriate to the arm size of children, who sat for at least 15 min before the examination. Blood pressure levels for gender by age and height percentile were calculated based on European Society of Hypertension guidelines [12]. BMI was calculated according to the WHO-developed formula, which is defined as weight in kilograms divided by the square of height in metres (kg/m^2^). The LMS method was used to calculate BMI standard deviation scores (SDS) for age according to a WHO reference (BMI z-score) [13]. Body composition was determined by bioelectrical impedance analysis (BIA). BIA measurements were conducted using TANITA MC-580 M S MDD, TANITA MC-780MA-N, and TANITA MC-780 P MA devices to measure fat mass (FM, kg) and fat-free mass (FFM, kg).

The fasting blood sample was performed to determine biochemical and hormonal parameters (participants remained fasting for at least 12 h). Fasting glucose, fasting insulin, low-density lipoprotein cholesterol (LDL-C), HDL-C, total cholesterol (TC), triglycerides (TG), uric acid, ALT, Aspartate aminotransferase (AST), Gamma-glutamyl transferase (GGT), and Alkaline phosphatase (ALP) levels were measured in each centre by using commercial, widely available methods. An oral glucose tolerance test (OGTT) was conducted in children above the age of 10 years or younger in whom puberty had begun. Prediabetes was defined as the presence of impaired fasting glucose (IFG; fasting glucose plasma concentration between 100 mg/dL to <126 mg/dL) or impaired glucose tolerance (IGT; at 2 h plasma glucose concentration in OGTT of 140 mg/dL to 199 mg/dL) [14]. Insulin resistance (IR) was evaluated according to the homeostasis model assessment [HOMA-IR score calculated by the equation fasting insulin (U/mL) × fasting glucose (mg/dL)/405], quantitative insulin sensitivity check index [QUICKI = 1/log (fasting insulin (U/mL)) + log (fasting glucose (mg/dL))], and triglyceride glucose index [TyG index = ln [TG (mg/dL)  ×  fasting glucose (mg/dL)/2]. The paediatric NAFLD fibrosis index (PNFI) was calculated using the original formulas proposed by Nobili et al. [15]. Laboratory analyses were performed using commercial laboratory methods. The analysis of biochemical parameters was conducted using enzymatic methods, while the analysis of hormonal parameters was carried out using chemiluminescent methods.

In all patients, abdominal ultrasonography (US) was performed. Liver ultrasound was performed in four centres by four different physicians using the following machines: MyLab X90, Philips Affiniti 70, Samsung HS60, and Samsung RS85 with a convex probe. The liver diameter was measured on the midclavicular axis from the upper margin to the lower margin of the liver and noted in millimetres (mm). Sonographic assessment of the presence of hepatosteatosis was based on a comparison of the echogenicity of the liver and kidney and the absence of the echogenic diaphragm and walls of intrahepatic vessels.

According to the new MAFLD criteria (mentioned previously), MAFLD was diagnosed in the presence of steatosis, evaluated by ultrasonography and/or persistent elevated ALT concentrations (>52 U/L for boys and >44 U/L for girls) [3].

Patients aged 10 years or more have been diagnosed with MetS according to the International Diabetes Federation (IDF) Consensus Statement from 2007 [16]. This includes abdominal obesity (measured by WC), which has been met in all of our patients, and two or more of the following factors: hypertriglyceridemia, low HDL-C, increased fasting glucose level, and hypertension, according to age groups.

For obese children under 10 years of age, we applied modified IDF criteria. MetS was diagnosed if the WC was ≥90th percentile (age and sex specific), according to Polish references for children and adolescents [17], and at least 2 of the following criteria were presented: 1.Triglycerides ≥ 130 mg/dL [18];2.HDL-C < 10th percentile (age and sex-specific) [18];3.SBP and/or DBP ≥ 90th percentile (age, gender, and height specific), according to blood pressure references for Polish children and adolescents [19];4.Fasting glucose ≥ 100 mg/dL or known T2DM.

### 2.2. Statistical Analyses

The statistical analyses were performed using Statistica 14.0 PL. We described quantitative variables with a mean ± standard deviation for normal or median ± interquartile range for skewed distribution and qualitative variables by frequency (%). The normal distribution of data were checked with Kolmogorov–Smirnov test and Shapiro–Wilk tests. Chi-square test was used for qualitative data comparison of groups, and the independent sample *t*-Student test and U Mann–Whitney test were used for quantitative data comparison of groups, as appropriate. Correlation analysis was performed using the Pearson correlation coefficient for normally distributed samples, Spearman correlation coefficient for non-normally distributed data, and Gamma correlation for non-normally distributed data with many tide ranks. An analysis of covariance (ANCOVA) was applied to adjust WC and liver AP diameter in MAFLD and non-MAFLD patients for sex and age. *p*-value < 0.05 was considered statistically significant.

A multivariate logistic regression analysis was conducted to identify independent predictors of MAFLD. The final model included the following predictors: ALT/AST > 1.5, fasting glucose > 100 mg/dL, waist circumference (per 10 cm), sex (male vs. female), GGTP (per 10 U/L), liver anterior–posterior (AP) diameter measured by ultrasound (per 1 mm), and elevated HOMA-IR.

## 3. Results

The study group consisted of 212 patients [114 girls (F) (53.8%), 98 boys (M) (46.2%)] with a mean age of 13.2 ± 3.1 years [range 3–18 years], with mean z-score BMI 3.84 ± 0.83. MAFLD was diagnosed in 125 (59.0%) patients (F: 60, M: 65). The MAFLD was diagnosed by sonographic assessment of hepatosteatosis in 118 (55.4%) patients (F: 56, M: 62), and due to elevated ALT concentration in seven patients (3.3%), without features of liver steatosis in the US exam. In the group with hyperechogenic liver, the ALT value was not elevated in 89 patients (71.2%). Among children below ten years of age (*n* = 22), we diagnosed MAFLD in 45% (*n* = 10) of cases.

Clinical characteristics and liver parameters in the MAFLD and non-MAFLD groups are presented in Table 1 and Table 2.

We have not noted any significant differences between children with MAFLD and children without MAFLD regarding age, birth weight, duration of pregnancy, or obesity duration time. MAFLD was observed significantly more often in boys than in girls (*p* = 0.04). WCs were significantly higher in the MAFLD compared to the non-MAFLD group (*p* < 0.01). After adjusting for age and sex, the significance changed to tendency (*p* = 0.07). No significant differences were noted in BMI z-score, as well as in the content of body fat and muscle mass between patients with MAFLD and those without it. In the MAFLD group, we observed higher liver anterior–posterior midclavicular diameter compared to patients in the non-MAFLD group (*p* < 0.001), even after adjustment for age and sex (*p* = 0.001).

Liver function tests revealed higher ALT, AST, and GGT concentrations in the group of patients with MAFLD compared to the patients without. No group difference was found for ALP. In the MAFLD group, significantly higher fasting and 120′ OGTT insulin and 120′ OGTT glucose levels, as well as IR indexes (TyG, HOMA-IR, Quicky) were observed compared to the non-MAFLD group (Table 3). IGT and increased HOMA-IR value (>4) [20] occurred more frequently in the group with MAFLD than in non-MAFLD patients (respectively, 16.5% vs. 6.5%, *p* = 0.04; 87.2% vs. 74.4%, *p* = 0.021). In the lipid profile of patients with MAFLD, we observed significantly higher concentrations of TG and TG/HDL ratio. However, the total cholesterol, LDL-C, and HDL-C levels did not differ significantly (Table 3).

MetS was found in 57.5% (*n* = 50) of non-MAFLD patients and 66.5% (*n* = 83) of MAFLD patients (*p* > 0.05). Moreover, when we analysed the subgroup of patients below the age of 10, we also observed a higher frequency of MetS in the patients with MAFLD compared to the non-MAFLD group (60% vs. 41.6%, *p* > 0.05). High fasting blood glucose was the only MetS component that was significantly different between MAFLD and non-MAFLD patients (*p* = 0.017) (Table 4). In the MAFLD group, a significantly higher number of patients with four or more components of the MetS were observed compared to patients without MAFLD (33.6% vs. 12.6%, *p* = 0.0004) (Figure 1). The number of components of the MetS was significantly positively correlated with MAFLD (r_γ_ = 0.28, *p* < 0.001).

The multivariate logistic regression model showed that ALT/AST > 1.5 was independently associated with MAFLD (OR = 19.79; 95% CI: 2.19–342.76; *p* = 0.018), fasting glucose > 100 mg/dL was strongly associated with MAFLD (OR = 44.53; 95% CI: 3.40–1378.95; *p* = 0.009), and liver AP diameter was a significant continuous predictor (OR per 1 mm increase = 1.07; 95% CI: 1.03–1.12; *p* = 0.003).

The PNFI value was significantly higher in patients with MAFLD (9.87 ± 0.33 vs. 9.66 ± 0.74, *p* = 0.016). Among patients with MAFLD, the PNFI value > 9 was observed in 97.5% of cases, compared to 88.2% among non-MAFLD patients (*p* = 0.007), and it was strongly correlated with MAFLD (r_γ_ = 0.68, *p* < 0.001). The PNFI value correlated positively with the number of MetS components (r_γ_ = 0.28, *p* = 0.0004).

## 4. Discussion

Given the children and adolescents with severe obesity and the new definition of fatty liver disease, we decided to assess the prevalence of MAFLD in the group of children with severe obesity in the Polish population and characterise clinical and biochemical metabolic risk factors in this group. MAFLD prevalence in our cohort of children and adolescents with severe obesity was 59%. A comparison of the two groups, MAFLD versus non-MAFLD, showed significant differences in the liver AP midclavicular diameter, whereas no differences in body mass (expressed through BMI z-score) and body composition have been observed. MAFLD was characterised by higher insulin levels, and all calculated IR indexes (HOMA-IR, Quicky, TyG index), but also the TG and TG/HDL-C ratio compared to the non-MAFLD group. In patients with MAFLD, more MetS components were observed than in the non-MAFLD group. We calculated PNFI, a non-invasive method to study liver fibrosis, and we observed no differences in PNFI between children with MAFLD and those without. Whether this parameter is suitable for assessing liver fibrosis in the group of children and adolescents with extreme obesity needs to be investigated in further studies.

The available data have been based on the definition of NAFLD, and the overall mean prevalence of NAFLD in children and adolescents with obesity was estimated at 34.2% [21]. So far, little has been reported in the literature about the prevalence of MAFLD in children and adolescents. For example, a Mexican study revealed MAFLD in 12.6% of children with overweight and obesity in elementary school [22]. In another meta-analysis from 2021, the authors revealed that the overall prevalence of MAFLD was 33.78% in the general population and 44.94% in paediatric patients with obesity [23]. There are ethnical and racial differences in MAFLD prevalence. A higher risk for MAFLD is reported in the Hispanic ethnic groups and obese Asian children [3,24], with the Caucasian population showing a midway prevalence (33%) [25]. In our study, the prevalence of MAFLD was 59.0%, nearly twice as high as in previous research. This discrepancy may be due to the specific group that was examined as being characterised by extreme obesity. According to our knowledge, our study is one of the first conducted exclusively on Caucasian children and adolescents with extreme obesity assessing the prevalence of MAFLD. We did not observe a difference in BMI z-score between the MAFLD and non-MAFLD groups. WC was significantly higher among patients with MAFLD than in the non-MAFLD group; however, the effect disappeared after age and sex were adjusted. An analysis in a larger cohort is necessary to investigate this aspect further.

IR is a well-documented key factor linking MAFLD and MetS in children and adolescents with obesity [26]. In our study, insulin levels, HOMA-IR, and Quicky were higher in children and adolescents from the MAFLD group than in children and adolescents from the non-MAFLD group. We also demonstrated that elevated HOMA-IR and impaired fasting glucose are among the strongest predictors of MAFLD. Previous studies [27,28] had shown similar findings that insulin and HOMA-IR were higher in groups with MAFLD. Interestingly, several studies have demonstrated that NAFLD is a major risk factor for IR and T2DM [29,30], which suggests that IR could be both a risk factor and a consequence of MAFLD. In our cohort, the TyG index was also significantly higher among children and adolescents with extreme obesity in the MAFLD group than in the non-MAFLD group. The TyG index, calculated from fasting glucose and TG levels, has been documented as an important indicator of IR and hepatic triglyceride content [31], which are major components in MAFLD pathogenesis. Moreover, previous studies suggested the TyG index is more reliable than insulin level and HOMA-IR, a predictor and diagnostic tool of hepatic steatosis [32,33]. Additionally, the TyG index was confirmed to be superior to HOMA-IR for T2DM detection [34]. However, TyG still does not have a well-established cut-off point for the diagnosis of significant hepatic steatosis.

In our study, we found similar findings in the lipid profile as in previous research performed in cohorts of children and adolescents with obesity [35,36,37]: higher TG levels and TG to HDL-C ratios among the MAFLD group compared to non-MAFLD group children and adolescents with obesity. In accordance with Bălănescu A. et al. [36], we have not observed significant differences in TC and LDL-C concentrations among the MAFLD and non-MAFLD groups. However, in comparison to them, we have not proved the significance of lower HDL-C levels among children with hepatosteatosis, as the HDL-C level was similar in both groups (*p* = 0.16). Jimenez-Rivera et al. reported a similar lipid profile in NAFLD children with obesity to we observed in our cohort [37]. On the other hand, there are differences in dyslipidemic patterns of MAFLD patients in several studies. Atabek et al. [38] found high TC with high LDL-C levels and similar levels of TG and HDL-C in NAFLD patients versus non-NAFLD patients with obesity. Nigam et al. [39] reported increased serum TC and TG levels and decreased HDL-C levels in adult NAFLD Asian Indian patients than in non-NAFLD cases. Resuming the most common type of dyslipidaemia in MAFLD is characterised by low HDL-C and high TG, which provide high risk cardiometabolic factors. The TG/HDL-C ratio might be a considerable cardiovascular risk marker induced by IR in paediatric patients with MAFLD.

The strong correlation between pathogenesis, morbidity NAFLD and MetS caused the redefinition of this condition as ‘Metabolic Dysfunction Associated Fatty Liver Disease’, highlighting hepatosteatosis as a crucial disorder strictly connected with clinical and laboratory abnormalities of MetS. We have recognised MetS following IDF criteria. Nevertheless, they are dedicated to children above ten years old. However, our patients characterise severe obesity connected with high cardiometabolic risk even in prepubertal age. Thus, we decided to implement modified IDF criteria for children from 3 years of age. In all MAFLD individuals (aged 3–18 years), MetS was recognised in 66.4% of patients and in 45.0% of the youngest MAFLD children aged 3–10 years. In the presented study, patients with MAFLD exhibited a higher prevalence of MetS than in previous studies [40,41,42]. Turkish study have demonstrated tenfold higher prevalence of MetS in children with hepatosteatosis (24.3%) than in those without [41]. Fu JF et al. revealed a three times higher prevalence MetS in NAFLD obese children than in non-NAFLD obese (39.74% vs. 12.04%, *p* < 0.05) [42]. This difference may arise from the high degree of obesity and larger WC of our patients compared to those in other studies. WC is an indicator of visceral obesity, which is a contributing factor to the development of MetS and cardiovascular disease [43]. Additionally, in our study, the frequency of almost all components of the MetS (without abdominal obesity), was higher in the MAFLD-group than in the non-MAFLD group. However, only higher fasting glucose level incidence was significantly higher in MAFLD than in non-MAFLD subjects. This indicates that MAFLD is a predictor of developing MetS and an increased risk of its components, especially glucose metabolism disorders. Therefore, patients diagnosed with MAFLD should be under special medical supervision for the detection and prevention of metabolic complications.

The complication of simple liver steatosis may be an irreversible process of fibrosis closely associated with cirrhosis and hepatocellular carcinoma (HCC). Hepatic fibrosis seems to be more aggressive in severely obese adolescents than in adults with high risks for liver-related mortality [44]. The prevalence of advanced fibrosis in children with NAFLD has been estimated at 10–20% [45]. Liver biopsy is the gold standard for diagnosing fibrosis in patients with NAFLD. However, that is an invasive, costly procedure that is difficult to perform in morbidly obese patients. Hence, simple, noninvasive tests have been developed to assess of liver fibrosis. PNFI, based on WC, TG levels, and age, is one of the predictive factors of paediatric liver fibrosis. According to Nobili V et al., the value ≥ 9 PNFI could be predictive for diagnosing liver fibrosis in children and adolescents. [15]. Mosca A. et al. in their research study, confirmed that the PNFI remains the best non-invasive score in paediatric age for NAFLD [46]. They diagnosed significant clinical liver fibrosis (F > 1) among 11.7% NAFLD adolescents; however, minimal fibrosis (F = 1) was revealed among 49% of participants. The PNFI value was the highest in the group with clinically significant fibrosis (F > 1): 7.5 (4–9.5), also Alkhouri N et al. [47] revealed significantly higher PNFI value among participants with F2-F3 fibrosis stadium in comparison with mild fibrosis (F0-F1) accordingly 8.8 ± 1.9 vs. 6.9 ± 2.7. In accordance with previous studies, we confirmed a significant difference in PNFI value between MAFLD and non-MAFLD groups (respectively: 9.87 ± 0.33 vs. 9.62 ± 0.74; *p* < 0.05). Interestingly, our patients have obtained a considerably higher value of this non-invasive fibrosis score; nearly all (97.5%) MAFLD patients have had PNFI > 9, which might suggest that nearly all have liver fibrosis, in disagreement with the previous data. Moreover, even in patients without MAFLD, PNFI > 9 was found among 88% of participants. Hence, that could be speculated that the PNFI is likely not to be a good predictive factor of hepatic fibrosis in children with severe obesity, which might be due to the very large waist circumference in this group influencing directly and significantly the value of the index. It appears that PNFI may be helpful for assessing the prediction of fibrosis up to a certain, yet undetermined, degree of abdominal obesity. However, in patients with severe obesity, its application may be misleading. On the other hand, this score is a good predictor for cardiometabolic risk in adolescents with overweight or obesity [34]. Furdela V et al. revealed that the maximum likelihood in the prediction of MetS is achieved by combining the lipid indexes TyG index, PNFI, and TG/HDL-C [34]. In accordance with that, we proved a positive correlation between the PNFI value and the number of MetS components. To sum up, yet we do not have insufficient data to assess the risk of liver fibrosis in children and adolescents with extreme obesity, but we have tools enabling the assessment of the risk of MetS prevalence.

## 5. Limitations of the Present Study

We are aware of the limitations of the study, the liver steatosis was diagnosed by assessing liver echogenicity based on US by doctors from four different clinical centres, which can lead to discrepancies in subjective assessment. However, US is currently an easy, non-invasive, and safe method for assessing hepatosteatosis. Another noninvasive parameter of liver tissue damage is Shear Wave Elastography (SWE). Nevertheless, there are contradictions in the disease-specific SWE cutoff, mean, and normal reference range. Most studies obtained higher SWE values in NASH obese and overweight patients. Nevertheless, the values are widely divergent [48]. This is a cross-sectional study involving only children and adolescents with severe obesity. As such, the study has certain limitations. To obtain more detailed and generalizable results, further research with a larger sample size is needed. Future studies should include patients with varying degrees of obesity, as well as a control group of individuals with normal weight, to rule out confounding factors associated with morbid obesity.

## 6. Conclusions

MAFLD prevalence in children with severe obesity is nearly twice as high as in children with overall obesity and reaches 59%. In this cohort, liver AP midclavicular diameter is the only differentiating anthropometric factor between patients with MAFLD and non-MAFLD. Moreover, MAFLD patients are characterised by more pronounced IR and hypertriglyceridemia, as well as a higher number of MetS components than non-MAFLD patients. It could be speculated that PNFI is likely not to be a good predictive factor of liver fibrosis in children and adolescents with severe obesity.

## Figures and Tables

**Figure 1 jcm-14-03565-f001:**
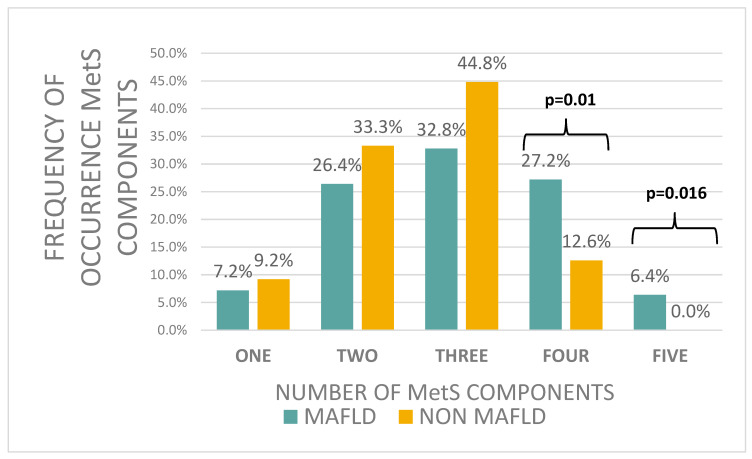
Frequency of occurrence MetS components.

**Table 1 jcm-14-03565-t001:** Clinical, anthropometric, and bioimpedance characteristics of the children and adolescents with MAFLD (*n* = 125) to children and adolescents without MAFLD (*n* = 87).

	Patients with MAFLD	Patients Without MAFLD	*p*-Value
Age [years]	14.0 ± 3.0	13 ± 3.4	0.2 **
Age of obesity onset [years]	4.5 ± 6.5	3.0 ± 8.0	0.52 **
Obesity duration time [years]	8.0 ± 7.0	8.0 ± 7.5	0.36 **
Birth weight [g]	3468.4 ± 508.5	3342.7 ± 612.7	0.06 *
BMI Z-SCORE	3.7 ± 0.5	3.6 ± 0.5	0.25 **
Waist circumference [cm]	115.2 ± 12.6	109.6 ± 15.2	0.004 * 0.07 ^#^
Fat mass [%]	46.1 ± 9.1	46.9 ± 8.1	0.71 **
Free fat mass [%]	53.8 ± 9.2	53.4 ± 9.0	0.93 **
Muscle mass [kg]	53.9 ± 17.6	49.7 ± 12.2	0.32 *

Data are shown as the mean ± standard deviation and Me ± intequartile range; * *t*-student test; ** U- Mann–Whitney test; ^#^—adjusted for age and sex; BMI: Body Mass Index.

**Table 2 jcm-14-03565-t002:** Comparison of concentrations of liver parameters in children and adolescents with MAFLD (*n* = 125) to children and adolescents without MAFLD (*n* = 87).

	Patients with MAFLD	Patients Without MAFLD	*p*-Value
Liver AP midclavicular diameter [mm]	149.4 ± 17.9	135.1 ± 17.5	<0.0001 * 0.001 ^#^
Alanine aminotransferase [μU/mL]	31.3 ± 23.6	24.0 ± 10.0	<0.0001 **
Aspartate aminotransferase [μU/mL]	26.4 ± 13.0	21.0 ± 13.0	<0.0001 **
ALT/AST ratio	1.2 ± 0.6	1.1 ± 0.4	0.007 **
Gamma-glutamyl transferase [μU/mL]	23.0 ± 14.0	18.0 ± 11.0	<0.0001 **
Alkaline fosfatase [μU/mL]	118.0 ± 123.8	141 ± 147	0.74 **
PNFI	9.87 ± 0.33	9.62 ± 0.74	0.016 **

Data are shown as the mean ± standard deviation and Me ± intequartile range; * *t*-student test; ** U- Mann–Whitney test; ^#^—adjusted for age and sex; PNFI: Paediatric NAFLD fibrosis index, AP anterior–posterior.

**Table 3 jcm-14-03565-t003:** Comparison of biochemical characteristics of the children and adolescents with MAFLD (*n* = 125) to children and adolescents without MAFLD (*n* = 87).

	Patients with MAFLD	Patients Without MAFLD	*p*-Value
Glucose 0 [mg/dL]	89 ± 13.0	87.0 ± 11.2	0.08 **
Glucose 120 [mg/dL]	118 ± 31.6	105.0 ± 30.0	0.0006 **
Insulin 0 [IU/mL]	27.9 ± 16.8	22.5 ± 16.3	0.0008 **
Insulin 120 [IU/mL]	135.1 ± 115.4	85.0 ± 92.1	0.004 **
HOMA-IR	6.1 ± 4.1	4.7 ± 4.2	0.0005 **
QUICKY	0.4 ± 0.1	0.4 ± 0.1	0.0006 **
TyG	8.7 ± 0.7	8.4 ± 0.5	0.002 **
Total cholesterol [mg/dL]	161.0 ± 47.8	159.1 ± 33.0	0.3 **
LDL cholesterol [mg/dL]	94.0 ± 40.0	93.0 ± 33.3	0.99 **
HDL cholesterol [mg/dL]	40.6 ± 7.6	42.2 ± 9.1	0.16 *
Triglycerides [mg/dL]	125.0 ± 81.9	107.5 ± 60.4	0.008 **
TG/HDL-C ratio	3.3 ± 2.3	2.6 ± 1.7	0.006 **
Uric Acid [umol/L]	404.5 ± 111.0	362.8 ± 93.8	0.0016 **
CRP [mg/L]	4.4 ± 3.9	5.0 ± 5.0	0.4 **

Data are shown as the mean ± standard deviation and Me ± intequartile range; * *t*-student test; ** U—Mann–Whitney test; HOMA-IR: Homeostatic Model Assessment for Insulin Resistance, CRP: C-reactive protein, QUICKY: Quantitative Insulin Sensitivity Check Index, TyG: triglyceride-glucose index.

**Table 4 jcm-14-03565-t004:** Comparison of the presence of parameters of the Metabolic Syndrome in children and adolescents with the MAFLD (*n* = 125) and without MAFLD (*n* = 87).

COMPONENT OF METABOLIC SYNDROME	Patients with MAFLD, *n* (%)	Patients Without MAFLD, *n* (%)	*p*-Value
ABDOMINAL OBESITY Waist Circumference ≥ 90th percentile	125 (100%)	87 (100%)	1.0
HIGH BLOOD PRESSURE ≥90th percentile <10 years; >10 years: Systolic BP ≥ 130 or diastolic BP ≥ 85 mmHg or treatment of previously diagnosed hypertension	105 (86.7%)	67 (80.7%)	0.24
HIGH FASTING GLUCOSE FPG ≥ 100 mg/dL or known T2DM	17 (13.8%)	3 (3.4%)	0.017
HYPERTRIGLICERYDEMIA ≥130 mg/dL below the age 10, ≥150 mg/dL above the age 10, or specific treatment for high Triglycerides	41 (32.8%)	19 (21.8%)	0.08
LOW HDL CHOLESTEROL <10th percentile for age and sex below age of 10 <40 mg/dL in males and females 10–16 years and <50 mg/dL in females ≥ 16 years, or specific treatment for low HDL cholesterol	70 (56.0%)	41 (47.1%)	0.2

## Data Availability

The datasets during and/or analysed during the current study available from the corresponding author on reasonable request.

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
