# Peer review of "High Prevalence of Metabolic-Associated Fatty Liver Disease (MAFLD) in Children and Adolescents with Severe Obesity"

_jcm, 2025, doi:10.3390/jcm14103565_

Round 1

Reviewer 1 Report

Comments and Suggestions for Authors

The authors must be congratulated on their work, which is of particular significance given the increasing prevalence of the disease in parallel with the increase in obesity. A plethora of studies have been conducted on the adult population; consequently, it is imperative that the target population in this particular case consists of children and adolescents residing within a specified geographical area.

The study is meticulously structured, with a clearly defined objective and adequate data collection, in addition to the statistical tests utilised.

The results obtained are presented in a clear and coherent manner, accompanied by a thorough discussion that engages with the extant literature on the subject.

As posited by the authors, the study is subject to certain limitations. Chief among these is the hepatic evaluation by ultrasound, which is contingent upon the proficiency of the operator.

Author Response

We wish to express our profound gratitude for the time and thoughtful consideration you dedicated to the review of our manuscript. We are truly honored by your positive evaluation and appreciative of your recognition of the significance of our research.

Authors

Reviewer 2 Report

Comments and Suggestions for Authors

I read with great interest the manuscript ID jcm-3602993 entitled “High prevalence of metabolic-associated fatty liver disease (MAFLD) in children and adolescents with severe obesity”. It tackles an important subject which is the frequency of MAFLD in children with obesity.

Title:

The term prevalence is not accurate, better to be replaced by frequency. Prevalence calculation needs much bigger sample sizes.

Methodology:

It would have bee nice to add those with moderate and mild obesity and contols as we don't have a cut off value for MAFLD in this age group (3 years). It would be nice to add this to the limitations.

The definition of obesity is not accurate given the wide age range, better to use the BMI precentiles to be more accurate.

It would be nice to add a refrence for the criteria for severe obesity definition.

For the body weight, height, BMI, waist and waist to hip ratio, better to use the percentiles and add refrences for them.

Please add refrence for the Tanner stage and blood pressure percentiles.

Please indicate the duration of fasting for biochemical samples.

It would be nice to add the methodology of biochemical tests done.

For the HOMA IR please add a refrence.

Please add the device and methodology of ultrasound and if it was done by single pediatric radiologist.

For the MAFLD diagnostic criteria, is the given ALT values valid for all age groups? I think it differs according to age.

Table 2 is of no value, since the compared parameters are already used to diagnose MAFLD.

It would be nice to add multivariate regression for factors assocaited with MAFLD.

Limitations:

It would be nice to add the cross sectional nature of the study.

Minor comments:

Better to avoid the terms we and our.

Author Response

Dear Reviewer,

Thank you very much for taking the time to review this manuscript. Please find the corresponding revisions in track changes in the re-submitted files.

For a research article

Response to Reviewer 2 Comments:

  1. Summary

Thank you very much for taking the time to review this manuscript. Please find the detailed responses below and the corresponding revisions in track changes in the re-submitted files. We  hope that it has once again significantly improved manuscript’s quality

  1. Point-by-point response to Comments and Suggestions for Authors

Comments 1: The term prevalence is not accurate, better to be replaced by frequency. Prevalence calculation needs much bigger sample sizes

Response 1: We thank the reviewer for that comment. It allowed us to clarify this aspect of the study. According to literature [1,2], “prevalence” or “incidence” should be used instead of the more general term “frequency”, depending on whether one refers to the total burden or the rate of new cases over time. Incidence in particular describes the risk of developing a disease within a defined period in a previously unaffected population. Prevalence refers to the proportion of all cases of a condition within a given population at a specific point in time. Taking into account that our study is cross-sectional in design, the use of the term “prevalence” seems more appropriate and aligns with standard epidemiological terminology.

  1. Dastgiri S. Frequency, incidence, and prevalence. Saudi Med J. 2016 Mar;37(3):329. doi: 10.15537/smj.2016.3.14677. PMID: 26905360; PMCID: PMC4800902.
  2. Noordzij M, Dekker FW, Zoccali C, Jager KJ. Measures of disease frequency: prevalence and incidence. Nephron Clin Pract. 2010;115(1):c17-20. doi: 10.1159/000286345. Epub 2010 Feb 19. PMID: 20173345

Comments 2:  It would have been nice to add those with moderate and mild obesity and contols as we don't have a cut off value for MAFLD in this age group (3 years). It would be nice to add this to the limitations

Response 2: The primary aim of our project was to investigate the genetic determinants of severe obesity (defined according to the literature [3], as in the inclusion criteria), as well as to characterise the associated biochemical profile and comorbidities in this population. We believe that future studies including individuals with moderate and mild obesity, as well as healthy controls, would provide valuable comparative insights and help to elucidate the full spectrum of metabolic alterations further.

We changed “limitation” section:

“This is a cross-sectional study involving only children and adolescents with severe obesity. As such, the study has certain limitations. To obtain more detailed and generalizable results, further research with a larger sample size is needed. Future studies should include patients with varying degrees of obesity, as well as a control group of individuals with normal weight, to rule out confounding factors associated with morbid obesity."

According to the European Society for Paediatric Gastroenterology, Hepatology and Nutrition (ESPGHAN) [4], the diagnosis of MAFLD in children under the age of 10 is infrequent, and it is considered exceptionally rare in children younger than 3 years. This presents a significant clinical challenge in recognizing MAFLD in this age group. Nevertheless, the diagnostic criteria for MAFLD in children and adolescents, as proposed in the 2021 consensus, can be applied from the age of 2 years onwards.

[9].von Schnurbein, J., Wabitsch, M. Monogene Adipositas. Medgen 29, 348–359 (2017). https://doi.org/10.1007/s11825-017-0157-z 

[5]. Vajro, P., Lenta, S., Socha, P., Dhawan, A., McKiernan, P., Baumann, U., Durmaz, O., Lacaille, F., McLin, V. and Nobili, V. (2012), Diagnosis of Nonalcoholic Fatty Liver Disease in Children and Adolescents. Journal of Pediatric Gastroenterology and Nutrition, 54: 700-713. 

[3]. Eslam M., Alkhouri N., Vajro P., et al. Defining paediatric metabolic (dysfunction)-associated fatty liver disease: an international expert consensus statement. Lancet Gastroenterol Hepatol. 2021;6(10): 864-873. doi: 10.1016/S2468-1253(21)00183-7. 

Comments 3: The definition of obesity is not accurate given the wide age range, better to use the BMI precentiles to be more accurate. It would be nice to add a refrence for the criteria for severe obesity definition.

Response 3: ​The definition of severe obesity in children varies between organizations:​

World Health Organization (WHO): For children and adolescents aged 5–19 years, obesity is defined as a BMI-for-age greater than +2 standard deviations (z-scores) above the WHO Growth Reference median. However, the WHO does not provide a specific definition for severe obesity in this age group [6]. ​
Centers for Disease Control and Prevention (CDC): For children and adolescents aged 2 to 20 years, severe obesity is defined as a BMI at or above 120% of the 95th percentile for age and sex, or a BMI of 35 kg/m² or greater, whichever is lower [7].

As stated in the manuscript, in research contexts, some studies have used a BMI-for-age z-score greater than +3 as a criterion for severe obesity in children. This approach is based on the WHO growth standards and is often employed in studies focusing on the most extreme cases of pediatric obesity [8]. However, in line with the primary objective of our study, we applied even more stringent inclusion criteria (to establish the Polish database of severely obese children and adolescents and to evaluate the prevalence of monogenic forms of obesity in this cohort”)[3]. All participants included in the analysis had a BMI z-score greater than +3. 

We added to manuscript bibliography in  BMI inclusion criteria according to von Schnurbein, and  J., Wabitsch proposition [9]. 

[9]. von Schnurbein, Julia and Wabitsch, Martin. "Monogene Adipositas: Pathophysiologie – Diagnostik – Therapieoptionen" Medizinische Genetik, vol. 29, no. 4, 2017, pp. 348-359.

  1. https://www.who.int/news-room/fact-sheets/detail/obesity-and-overweight 
  2. Prevalence of Overweight, Obesity, and Severe Obesity Among Children and Adolescents Aged 2–19 Years: United States, 1963–1965 Through 2015–2016 by Cheryl D. Fryar, M.S.P.H., Margaret D. Carroll, M.S.P.H., and Cynthia L. Ogden, Ph.D., Division of Health and Nutrition Examination Surveys. NATIONAL CENTER FOR HEALTH STATISTICS; Health E-Stats, Sep 2018

[2] Spinelli A, Buoncristiano M, Kovacs VA, et al. Prevalence of Severe Obesity among Primary School Children in 21 European Countries. Obes Facts. 2019;12(2):244-258. doi: 10.1159/000500436. Epub 2019 Apr 26. PMID: 31030201; PMCID: PMC6547273.

Comments 4: For the body weight, height, BMI, waist and waist to hip ratio, better to use the percentiles and add refrences for them.

Response 4: All of our participants met the inclusion criteria for both body weight and BMI above the 97th percentile. In order to more precisely illustrate the degree of obesity in our cohort—well exceeding 120% of the 95th percentile—we calculated BMI z-scores according to the WHO reference standards. In our view, this provides readers with a clearer basis for comparing the severity of obesity across different studies. Additionally, as shown in Table 4, each participant had a waist circumference above the 90th percentile according to  references for waist and hip circumferences in Polish children and adolescents 3-18 year of age [9].  We put this charts into bibliography. 

[16]. . Kotowska A., Gurzkowska B., Góźdź M.,et al. References for waist and hip circumferences in Polish children and adolescents 3-18 year of age. Standardy Medyczne Pediatria. 2015;1:137-150

Comments 5: Please add refrence for the Tanner stage and blood pressure percentiles.

Response 5: We have completed reference for Tanner stage and added two positions to bibliography:

“[10]. Marshall WA, Tanner JM. Variations in the pattern of pubertal changes in boys. Arch Dis Child. 1970;45(239):13-23.

[11]. Marshall WA, Tanner JM. Variations in pattern of pubertal changes in girls. Arch Dis Child. 1969;44(235):291-303.” 

The references for blood pressure given in manuscrpit:. 

“[19]. Krzyzaniak A., KrzywiÅ„ska-Wiewiorowska M., StawiÅ„ska-WitoszyÅ„ska B.,et al. Blood pressure references for Polish children and adolescents. Eur J Pediatr. 2009;168(11):1335-42. doi: 10.1007/s00431-009-0931-2.”

Comments 6: Please indicate the duration of fasting for biochemical samples.

Response 6: We added in section Materials and Methods”

“The fasting blood sample was performed to determine biochemical and hormonal parameters (participants remained fasting for at least 12 hours). “

Comments 7:  It would be nice to add the methodology of biochemical tests done.

Response 7: We  added:

Laboratory analysis was performed using commercial laboratory methods. The analysis of biochemical parameters was conducted using enzymatic methods, while the analysis of hormonal parameters was carried out using chemiluminescent methods.”

Comments 8: For the HOMA IR please add a refrence.

Response 8:

 I sincerely appreciate your bringing this matter to my attention. However, we did not use HOMA-IR charts because, despite studies that have attempted to establish normal HOMA-IR values for children and adolescents, reliable percentile charts are not yet available. However, based on selected percentile grids for HOMA-IR in overweight Caucasian children [10], we adopted a cut-off point for insulin resistance at HOMA-IR > 4, as this value typically corresponds to the 90th percentile on HOMA-IR charts adjusted for sex and age. We described in manuscript that HOMA- IR >4 is increased. We have completed the missing bibliography- no 19 in manuscript

[20]. Shashaj B., Luciano R., Contoli B., Morino G.S., Spreghini M.R., Rustico C., Sforza R.W., Dallapiccola B., Manco M. Reference ranges of HOMA-IR in normal-weight and obese young caucasians. Acta Diabetol. 2016;53:251–260. doi: 10.1007/s00592-015-0782-4. 

Comments 9: Please add the device and methodology of ultrasound and if it was done by single pediatric radiologist.

Response 9: We added the sentence with description of device and physicians. 

In all patients, abdominal ultrasonography (US) was performed. Liver ultrasound was performed in four centers by four different physicians using the following machines: MyLab X90, PHILIPS Affiniti 70, and Samsung HS60, Samsung RS85 with a convex probe.

In the Limitations section, we highlighted, “We are aware of the limitations of the study. The liver steatosis was diagnosed by assessing liver echogenicity based on ultrasound by doctors from four different clinical centres, which can lead to discrepancies in subjective assessment.” 

Comments 10: For the MAFLD diagnostic criteria, is the given ALT values valid for all age groups? I think it differs according to age.

Response 10: Thank you for your valid point; indeed, ALT values do vary slightly depending on age, particularly in the first year of life. However, the authors of the 2021 MAFLD criteria suggested: ”For detection of steatosis, we recommend liver imaging with ultrasound, controlled attenuation parameter, or persistent elevated ALT concentrations to more than twice the upper limit of normal (<26 U/L for boys and <22 U/L for girls).” [3]

[3] Eslam M., Alkhouri N., Vajro P., et al. Defining paediatric metabolic (dysfunction)-associated fatty liver disease: an international expert consensus statement. Lancet Gastroenterol Hepatol. 2021;6(10): 864-873. doi: 10.1016/S2468-1253(21)00183-7.

Comments 11: Table 2 is of no value, since the compared parameters are already used to diagnose MAFLD.

Response 11: In Table 2, we included what we believe to be important information that shows practicing physicians which easily accessible diagnostic methods (such as ultrasound and selected biochemical tests) are particularly relevant in diagnosing MAFLD, as these parameters are significantly elevated in patients with MAFLD. Additionally, we included information regarding the difference in the mean PNFI values (between the MAFLD and non-MAFLD groups), which we discuss in the 'Discussion' section. We would very much like to keep this table; however, if you still consider it to be irrelevant, we will of course remove it

Comments 12: It would be nice to add multivariate regression for factors assocaited with MAFLD.

Response 12: We would like to express our sincere gratitude for the valuable suggestion, which may enhance the scientific quality of our work. We performed a multivariate regression analysis. The methodology, results, and figure have been included in the text.

“A multivariate logistic regression analysis was conducted to identify independent predictors of MAFLD. . The final model included the following predictors: ALT/AST > 1.5, fasting glucose > 100 mg/dL, waist circumference (per 10 cm), sex (male vs female), GGTP (per 10 U/L), liver anterior-posterior (AP) diameter measured by ultrasound (per 1 mm) and elevated HOMA-IR. “

The multivariate logistic regression model showed that: ALT/AST > 1.5 was independently associated with MAFLD (OR = 19.79; 95% CI: 2.19–342.76; p = 0.018), Fasting glucose > 100 mg/dL was strongly associated with MAFLD (OR = 44.53; 95% CI: 3.40–1378.95; p = 0.009), Liver AP diameter was a significant continuous .”

Comments 13: Limitations: It would be nice to add the cross sectional nature of the study.

 Response 13: We have modified Limitation section to emphasise this point.

“This is a cross-sectional study involving only children and adolescents with severe obesity. As such, the study has certain limitations. To obtain more detailed and generalizable results, further research with a larger sample size is needed. Future studies should include patients with varying degrees of obesity, as well as a control group of individuals with normal weight, to rule out confounding factors associated with morbid obesity."

Comments 14: Minor comments: Better to avoid the terms we and our.

Response 14:  Thank You very much for this valuable indication.

Best regards

Round 2

Reviewer 2 Report

Comments and Suggestions for Authors

All comments were addressed properly.